# "It's a habit. They've been doing it for decades and they feel good and safe.": A qualitative study of barriers and opportunities to changing antimicrobial use in the Indonesian poultry sector

Rebecca Hibbard[1,2]*, Lorraine Chapot[1,2], Havan Yusuf[3], Kurnia Bagus Ariyanto[3], Kusnul Yuli Maulana[3], Widya Febriyani[3], Angus Cameron[1], Timothée Vergne[2], Céline Faverjon[1], Mathilde C. Paul[2]

**1** Ausvet Europe, Lyon, France, **2** INRAE, IHAP, ENVT, Université de Toulouse, Toulouse, France, **3** Ausvet, Jakarta, Indonesia

* rebecca.hibbard@envt.fr

**Data Availability Statement:** The data generated and analysed during this study are the interview

## Abstract

Interventions to change antimicrobial use (AMU) practices can help mitigate the risk of antimicrobial resistance (AMR) development. However, changing AMU practices can be challenging due to the complex nature of the factors influencing AMU-related behaviours. This study used a qualitative approach to explore the factors that influenced decision-making on AMU by farmers and other actors in the Indonesian poultry sector. Thirty-five semi-structured interviews were conducted with farmers, technical services staff from the private sector, and representatives of associations, universities, and international organisations in Central Java, West Java, and East Java. Thematic analysis identified three patterns of influence on AMU: how farmers used information to make AMU-related decisions, the importance of farmers' social and advisory networks, and the motivations driving changes in AMU behaviours. Key barriers identified included a lack of shared understanding around when to use antibiotics, financial pressures in the poultry sector, and a lack of engagement with government veterinary services. Potential opportunities identified included high farmer awareness of AMU, identification of private sector actors and peer networks as the stakeholders with established relationships of trust with farmers, and the importance of farmers' conceptions of good farming practices, which could be engaged with to improve AMU practices.

## Introduction

The use of antimicrobials in food-producing animals is of increasing concern in light of its potential contribution to the growing global threat of antimicrobial resistance (AMR). Southeast Asia has been identified as a hotspot for antimicrobial use (AMU) and AMR in both humans and food-producing animals, with AMU in this region projected to increase

transcripts, some of which contain personal information. The complete data are not publicly available for reasons of confidentiality, however, excerpts of the data to support the findings are included in the paper and in the Supporting Information. Anonymised and de-identified data may be requested from the corresponding author or Catriona Mackenzie, Director General at Ausvet Europe (catriona.mackenzie@ausvet.eu).

**Funding:** This research was funded by United States Agency for International Development (USAID) as part of the TRANSFORM (Transformational Strategies for Farm Output Risk Mitigation) Project. Grant number: 7200AA21CA00004. Website: https://www.usaid.gov/. The funders had no role in study design, data collection and analysis, decision to publish, or preparation of the manuscript.

**Competing interests:** The authors have declared that no competing interests exist.

in coming years [1, 2]. The poultry sector in particular has been identified as one of the sectors with the highest AMU globally, and rapid growth in this sector has the potential to drive further increases in AMU [3, 4]. High levels of AMU for prophylaxis and growth promotion have been reported on poultry farms in Asia [5–8], presenting a risk for the further development of AMR.

In Indonesia, high levels of AMU have been detected on broiler farms, with frequent overdosing and underdosing relative to recommended dosages, and high levels of prophylactic AMU alongside lower use for growth promotion, although the boundaries used by farmers to distinguish between AMU for treatment, prophylaxis, or growth promotion are often unclear [9, 10]. The presence of disease, farmer aspiration to prevent disease, and desire for improvement in productivity and growth have been identified as motivators for AMU among Indonesian broiler farmers, with contract farmers more likely than independent farmers to use AMs to prevent disease [11]. However, most research to date has concentrated on broiler farmers contracted to large integrated companies, with less data available on layer farmers (who tend to be independent farmers in Indonesia). In light of reports of high levels of multi-drug resistant bacteria in bacterial isolates from live poultry and chicken meat throughout Indonesia [12, 13], efforts to address AMR are of increasing importance in the country and the Southeast Asia region more broadly.

Interventions that promote responsible and prudent use of antimicrobials have the potential to mitigate the risk of AMR development [14]. However, interventions and policies that target farmers are unlikely to be successful without prior research or consultation to understand farmers' value framework and the contextual conditions within which they make decisions [15]. Effecting a change in AMU-related behaviours is particularly challenging due to the complexity of the factors which underpin and influence decisions around AMU, including personal behaviour and beliefs, farm management, and external factors outside of farmers' control [16]. The influence of other actors is also of importance. Farmers' intentions to reduce AMU are correlated with a perception that this will be met with approval by their social and advisory network [17], and the establishment of relationships based on mutual trust and understanding between farmers and technical support staff has been found to facilitate the provision of advice, support, and shared decision-making on AMU [18]. There is therefore a need for information to improve understanding of how these factors impact AMU decision-making by different actors in different contexts, to assist in shaping policy to change AMU-related behaviours [19, 20].

Qualitative research approaches can be particularly useful for gaining a contextualised understanding of individuals' experiences and how they make sense of their reality [21]. Such approaches can therefore be applied to improve understanding of how farmers and other actors make sense of information and decisions related to AMU within a specific context. There is an abundance of quantitative research on AMU-related behaviours of farmers and veterinarians in low- and middle-income countries (LMICs) [22–26]. Qualitative approaches have been less commonly used [27–29], despite their potential to elucidate the reasoning behind individuals' AMU practices and to identify barriers and drivers for behavioural change [30].

The objective of this paper is to explore the factors influencing AMU practices in the Indonesian poultry sector to identify the barriers and opportunities to effecting behavioural change in AMU. To do so, we used a qualitative approach to analyse farmers' and other stakeholders' views on the key factors influencing AMU and the challenges to improving AMU practices in poultry production. This study focuses on Indonesia as it is one of the countries with the greatest projected increase in AMU in food animals by 2030 [31], and the factors influencing AMU in the Indonesian poultry sector remain poorly understood. In addition, the Indonesian poultry industry has recently seen a dramatic decline in the number of independent farmers, as the

industry shifts towards greater integration and intensive farming systems [6, 32]. Similar changes are occurring in other Southeast Asian countries but their influence on AMU is still unknown. A better understanding of AMU behaviour in Indonesia may provide insight into potential approaches applicable across similar animal production systems throughout the region.

## Methodology

This research is reported in line with the Standards for Reporting Qualitative Research (SRQR) [33]. The completed SRQR checklist is provided in S1 Appendix. Although data collection and analysis are detailed in separate paragraphs for clarity, these were simultaneous and iterative processes.

### Qualitative approach and framework

This study was undertaken using an interpretivist approach, in which a focus is placed on participants' descriptions and understandings of their own experiences, and researcher subjectivity is a resource which influences data interpretation [34]. A wide range of different stakeholders were consulted to ensure a diversity of views, identified based on a stakeholder analysis of the Indonesian poultry sector. A semi-structured interview method was used to provide the opportunity for participants to introduce topics and issues of interest. The analytic framework used for this study borrowed aspects of constructivist grounded theory as espoused by Charmaz [35] (in particular, incorporating the view that both the data and the resulting analyses are constructed from the experiences of the participants and researchers, and basing the results on the data itself). This meant examining factors influencing AMU decision-making from the perspectives of different individuals, and grounding the analytical findings in the participants' experiences as they described them, while acknowledging the role of the researcher in interpreting these descriptions. Codes and themes were determined from and grounded in the data itself, rather than being predetermined based on the literature. This facilitated identification and analysis of unexpected themes in participants' responses.

### Study context

The Indonesian poultry industry consists of a commercial broiler sector, a commercial layer sector, and village households farming native chicken species [36], all of which may have different factors influencing farmers' AMU. This study focused principally on the layer sector, although farmers and other stakeholders from the broiler sector were also consulted. There are important differences between the layer and broiler sectors, most notably the predominance of contracted farms in the broiler sector (60–70% of farms are contracted to large integrator companies) compared to the layer sector in which most farmers are independent [36]. Additionally, layer farms have a relatively long production cycle (chickens are harvested at approximately 72 weeks) compared to broiler farms, which has implications for disease management and surveillance [36]. The region with the highest level of egg production and the highest concentration of layer farmers is East Java, followed by Central Java and West Java [36].

### Participants and sampling strategy

Participants were identified and prioritised through a stakeholder analysis of important actors in the Indonesian poultry sector. Based on the analysis, targeted participants included independent poultry farmers, contracted poultry farm managers (employed within integrated poultry systems), technical services (veterinarians and other technical staff employed by

private companies), as well as representatives of associations, private companies, academic institutions, and international organisations. A broad range of stakeholders were included to help ensure a diversity of views on the same topic would be heard. West Java, East Java, and Central Java were selected as the target regions for inclusion considering the high density of poultry farms in these regions. Participants were recruited between 20 June 2022 and 22 July 2022. Initially, a purposive sampling approach was used, with individuals invited to participate based on their expertise or key role within the poultry sector. Snowball sampling was subsequently used, with interviewees asked to suggest other relevant actors.

The sample size was determined both pragmatically and with regards to the concept of information power, whereby sample size is considered within the context of the study aim, participant specificity, use (or not) of established theory, quality of dialogue, and analysis strategy [37]. A relatively large sample size (for qualitative research, where sample sizes can range from one to 50 interviews depending on the study methodology [38–40]), was used based on the broad, exploratory nature of the study aim and analysis strategy, the sample specificity (the target to include multiple stakeholders from several different groups of stakeholders), and the highly variable quality of dialogue identified during data collection. An initial sample size of 20 was targeted based on the number of different stakeholder categories identified in the stakeholder analysis, assuming that multiple interviewees would be included within each category, and considering the number of interviewees that could reasonably be met with in the timeframe allocated for the study. New participants were gradually included until it was judged that a sufficient diversity of views was collected–when it was considered that the data collected was sufficiently rich to allow identification and analysis of the factors influencing AMU.

## Ethics approval

Ethics approval was obtained for the study (Medical and Health Research Ethics Committee, Faculty of Medicine, Public Health and Nursing, Universitas Gadjah Mada, Ref: KE/FK/0675/ EC/2022), and the methods approved as meeting the ethical principles outlined in the International and National Guidelines on ethical standards and procedures for research with human beings. Informed consent to participate and record the interviews was obtained verbally from all interviewees before their participation, and in writing where possible (for six participants).

## Data collection

**Data collection instruments–interview guide.**   A semi-structured approach was selected to ensure that information on the topics of interest was collected, while allowing participants to introduce topics which were important to them. Six different versions of an interview guide were developed for six different categories of interviewees (an example is provided in S2 Appendix). Four broad themes of questions were consistent across all versions of the interview guide: *transversal topics* (general questions about the interviewee's role and the main challenges they face), *disease prevention and control*, *AMU and stewardship*, and *health information and monitoring*, and were used as points of departure for forming interview questions and responding to interviewees. The interview guide was revised by all authors before being piloted internally with five of the authors, followed by an external stakeholder (an Indonesian poultry farmer). At each stage of piloting, the interview guide was revised.

**Data collection method–interviews.**   Interviews were conducted between 27 June 2022 and 22 July 2022. The interview team consisted of two international researchers and nine Indonesian personnel (three researchers, one intern, and five translators), with two to five interviewers present at each interview. Interviews were conducted in-person where possible but when not feasible an online video conferencing tool was used. In-person interviews were

carried out at a site chosen by the interviewee, usually their place of work (e.g., farm office for poultry farmers) or a neutral site (e.g., a cafe). Interviews were led by one of the Indonesian-speaking researchers to facilitate communication and build trust with the participants. Simultaneous translation was provided for the English-speaking researchers to facilitate note taking and familiarisation with the interview content, and to allow additional follow-up questions to be asked at the end of the interview. Interviews lasted from 60–180 minutes. The interview guide was used as the starting point for the interview, but where new topics were introduced by a participant, these topics were pursued according to interviewee engagement.

## Data processing

Thirty-five interviews were conducted. Thirty of the interviews were conducted in Indonesian–of these, 25 were recorded in the original language (Indonesian) and five were recorded in the English translation. Five interviews were conducted and recorded in English. All interviews were transcribed verbatim and anonymised using codes for each stakeholder. The Indonesian transcripts were subsequently translated to English by external professional translators and verified by the Indonesian-speaking authors against the original transcripts and audio recordings.

## Data analysis

**General approach.**   The dataset consisted of notes taken in the field during the interviews and the interview transcripts. Reflexive thematic analysis following the method of Braun and Clarke [41, 42] was employed, whereby the researcher plays an active role in engaging with and generating themes from the data to identify patterns of shared meaning. The interpretivist framework applied meant that there was a focus on the interviewees' descriptions of their experiences related to AMU and their explanations of these experiences. Analysis of the interview notes from all authors and transcripts was performed by the first author using NVivo [43]. Efforts to assure the integrity and trustworthiness of data were made through inclusion and consideration of material from all categories of stakeholders, to ensure a diversity of perspectives on the same topic was examined.

**Data familiarisation and data coding.**   Data familiarisation began during the interviews and continued throughout data processing and analysis. The interview notes and transcripts were read in full for quality control and familiarisation, with the Indonesian-speaking authors providing explanations and additional context as required. Initial coding was performed with codes defined broadly to identify excerpts of the notes and transcripts of potential relevance to the research question. Focused coding of the transcripts was then conducted to select and revise the most significant initial codes.

**Developing, reviewing, and defining themes.**   Themes were constructed from identification of shared meanings across the coded data that were underpinned by a central concept [41]. Codes were grouped together in increasing levels of abstraction and thematic maps were drawn to identify patterns across the codes. These were used as the basis for developing initial themes, which were subsequently refined after repeated readings of the transcripts and discussions with the other authors.

## Results and analysis

## Study participants

Thirty-five semi-structured individual and group interviews were conducted with 35 different individuals or groups (Table 1).

## Theme 1: How farmers negotiate the information available to them to make decisions on antimicrobial use

Almost all interviewees indicated during interviews that they routinely used antibiotics or recommended their use, with decisions based on the information directly available to them at the time.

**Personal knowledge and experience of identifying disease.** Interviewees said that in the absence of readily available diagnostic testing, the decision to use antibiotics was sometimes based on trial and error, broad categories of symptoms, and their own past experience.

*We are not used to laboratories because we have to act fast to do the treatment. So if we have done it many times, our risk of making a treatment error is minimal. (OTHER_10)*

However, interviewees recognised that reliance on past experience could act as a barrier to changing behaviour, where this was used to the exclusion of other information.

*He has been using antibiotics for decades, so changing to not using antibiotics is difficult [...] Because it's a habit. They've been doing it [using antibiotics] for decades and they feel good and safe. if you want to change it, it's difficult, I think. (FARMER_02)*

**Production parameters.** Several farmers had set thresholds for "severity" of a disease which they used to decide when and how to use antibiotics or to consult for treatment advice. Severity was measured in terms of mortality, morbidity, or reduced production. However, the thresholds at which action was taken, and the way in which farmers interpreted this information, varied greatly (Table 2). For example, some interviewees would use injections when only a few birds were affected, moving to mass in-water administration if the disease spread, while other interviewees only used injections if mass medication failed to resolve the problem.

**Situational (perceived as risk-based) decision-making.** Farmers' AMU was often situational, based on specific production stages identified as being at particular risk of disease. Most farmers referred to this preventive use as treatment, as they considered disease to be inevitable in the absence of antibiotics. The term "cleaning" was used by interviewees to refer to a short period (three to five days) of prophylactic AMU for different reasons.

Two specific circumstances when prophylactic AMU was considered unavoidable were identified. The first was in day-old chicks (DOCs) upon their arrival or within the first few days of arrival (mentioned by MANAGER_02, OTHER_09, OTHER_11, MANAGER_04, OTHER_14).

*The DOC from the factory does not guarantee that the DOC is healthy [...] So we need antibiotics at the beginning. (OTHER_14)*

The second circumstance was pre- or post-vaccination. Antibiotics were seen as necessary to prevent clinical signs subsequent to vaccination (mentioned by MANAGER_03, OTHER_06, OTHER_11, OTHER_13, OTHER_14).

*The vaccine first, then a few days later antibiotics. There is a post-vaccine reaction. (OTHER_11)*

Two farmers also simply referred to a "cleaning period" where chickens were routinely given antibiotics.

**Table 1. List of participating interviewees by category\*.**

| ID | Category | Role | Gender | Size of farms | Location of interview |
|---|---|---|---|---|---|
| ASSOCIATION_01 | Association | Secretary general | Male | N/A | Jakarta |
| ASSOCIATION_02** | Association | Secretary general | Male | N/A | Online |
| ASSOCIATION_03 | Association | Secretary general, chairmen/women | Female | N/A | Online |
| ASSOCIATION_04 | Association | Head of association | Male | N/A | Online |
| FARMER_01 | Independent layer farm | Farm worker | Female | 25,000 chickens | Bogor |
| FARMER_02 | Pullet farm | Farm owner/qualified veterinarian | Male | 6,000 chickens per house | Yogyakarta |
| FARMER_03 | Independent layer and pullet farm, feed mixer | Farm owner | Male | 40,000 chickens (all layers) | Yogyakarta |
| FARMER_04 | Independent layer farm | Farm owner | Male | 48,000 chickens (capacity for 70,000) | Malang |
| FARMER_05 | Independent layer and pullet farm, feed seller | Farm owner | Male | 4,500 chickens (all layers, capacity for 11,000) | Malang |
| FARMER_06** | Independent layer farm | Farm owner | Male | Information not provided | Yogyakarta |
| FARMER_07 | Independent layer farm | Farm manager | Male | 170,000 chickens | Karanganyar |
| FARMER_08 | Independent layer farm | Farm manager | Male | 65,000 chickens (60,000 layers but capacity for 100,000, 5,000 pullets) | Karanganyar |
| FARMER_09 | Independent layer and pullet farm | Farm owner/qualified veterinarian | Male | 70,000 chickens | Blitar |
| FARMER_10*** | Independent layer farm | Farm owner | Male | 14,000 chickens | Solo |
| FARMER_11 | Independent layer farm | Farm owner | Male | 50,000 chickens | Blitar |
| FARMER_12 | Independent layer farm | Farm owner | Male | 115,000 chickens (110,000 layers, 15,000 pullets) | Yogyakarta |
| MANAGER_01 | Integrator (breeding) | Assistant manager | Male | 5,000–10,000 chickens (broiler farms), 5,000–200,000 chickens (layer farms) | Jakarta |
| MANAGER_02 | Integrator (broiler) | Area manager | Female | 10,000–30,000 chickens | Online |
| MANAGER_03 | Integrator (broiler) | Farm supervisor | Male | 150,000 chickens (across three farms) | Solo |
| MANAGER_04 | Integrator | Head of farm unit | Male | 320,000 chickens (across several locations) | Solo |
| OTHER_01 | Pharmaceutical company | Commissioner | Male | N/A | Online |
| OTHER_02 | University | Professor | Female | N/A | Bogor |
| OTHER_04 | Feed company | Business manager | Male | Information not provided | Jakarta |
| OTHER_05 | International organisation | Team leaders | Male | N/A | Jakarta |
| OTHER_06 | Pharmaceutical company | Marketing representative | Male | Information not provided | Malang |
| OTHER_07 | Integrator | Company veterinarian | Male | 2000 chickens per house (18 houses) | Malang |
| OTHER_08 | Integrator | Technical services | Male | Information not provided | Malang |
| OTHER_09 | University | Professor | Male | N/A | Yogyakarta |
| OTHER_10 | Poultry shop supplying layer farms | Poultry shop owner/pullet farmer | Male | 10,000 chickens (pullets) per house (across four farms) | Blitar |
| OTHER_11 | Pharmaceutical company | Technical services | Female | N/A | Blitar |
| OTHER_12*** | Pharmaceutical company | Technical services | Female | N/A | Solo |
| OTHER_13 | Integrator (semi-integrated) (breeding, layer) | Assistant manager/company veterinarian | Male | 28,000–30,000 chickens (open farms), 26,000–40,000 chickens (closed farms) | Blitar |
| OTHER_14 | Pharmaceutical company | Technical services | Male | N/A | Yogyakarta |
| OTHER_15 | International organisation | Project leader | Male | N/A | Jakarta |
| OTHER_16 | Research institution | Researcher | Male | N/A | Online |

(*Continued*)

**Table 1.** (Continued)

| ID | Category | Role | Gender | Size of farms | Location of interview |
|---|---|---|---|---|---|
| OTHER_17 | University | Professor | Male | N/A | Online |

\* There is no OTHER_03 as this interview was not recorded and was therefore excluded from the results.

\*\* FARMER_06 is the same interviewee as ASSOCIATION_04 who was interviewed on two separate occasions, in their capacity as a farm owner, and as the head of an association respectively.

\*\*\* FARMER_10 and OTHER_12 were interviewed in the same interview.

> *In Indonesia there is a cleaning program called cleaning antibiotic. So at least for one month there must be three to five days of taking antibiotics with or without symptoms. (FARMER_02)*

> *For antibiotics, we take turns, every month we definitely have cleaning. (FARMER_08)*

The only consistent factors across these different definitions of "cleaning", were the short duration of treatment and its pre-emptive use in the absence of disease.

Interviewees considered AMU only within these circumstances to be a sign of good AMU management. A broiler farm manager (MANAGER_04) told us that their company's Standard Operating Procedures (SOPs) only permitted AMU within the first seven days of production, giving this as evidence of their responsibility in observing good AMU practices. Similarly, a technical services officer helping farmers to produce antibiotic-free broilers told us that good management meant only using antibiotics in DOCs.

## Theme 2: The influence of farmers' social and advisory networks

**The importance of the private sector and peers in farmers' AMU decisions.** Farmers usually engaged with other actors to inform their decision-making around AMU and disease management. They identified technical services (veterinarians and other technical staff employed by private companies) and peers as trusted sources of information.

The layer farmers interviewed received veterinary support through technical services from pharmaceutical companies, feed companies, and poultry shops.

**Table 2. Production parameters used to inform antibiotic use decisions and the associated actions taken by interviewees.**

| Production parameter | Threshold | Action taken if threshold reached | Interviewee ID* |
|---|---|---|---|
| Mortality | 1% | No action specified, but considered problematic | MANAGER_04 |
| | 4% | No action specified, but losses expected | OTHER_02 |
| Morbidity | 5% | Administer antibiotics in drinking water | FARMER_02 |
| | | Address farm management | |
| | 50% | Mass injection with antibiotics | FARMER_02 |
| | 10% | Under 10%: Administer antibiotics via drinking water | FARMER_03 |
| | | Over 10%: Administer antibiotics via mass injections | |
| | 20–30% | Previously, the farmer would treat orally with antibiotics when this threshold was reached. Now all antibiotics are given by injection regardless of number of birds affected. | FARMER_06 |
| Drop in production | 1–2% | Call the vet | FARMER_04 |

\* Please refer to Table 1 for information on the interviewees referred to by their ID in this table.

*We discuss it [disease] with colleagues who are the technical services, the experts in our perspective [. . .] And we usually also buy products from them. (FARMER_07)*

Broiler farmers and managers received advice from the technical services employed by the company to which they were contracted. They were provided with company guidance on how and when antibiotics should be administered, although final decisions on AMU were made by the farmer.

*When the chicken is sick, we will still treat it with antibiotics. But to use it, there are several steps. [. . .] you can first report to the technical services or its officers. (MANAGER_04)*

The technical services we interviewed told us that their role extended beyond selling products to providing advice and support for disease control and farm management.

*Like a salesperson, we sell the product of the company, and we also help the technical maintenance of the farm. So, we are not only and purely selling the product but also approaching the customer by helping the customer's problem. (OTHER_12)*

However, some farmers were wary of the potential influence of financial incentives for technical services staff to meet sales targets. These farmers were unlikely to purchase products or follow advice from someone who seemed to them to be prioritising sales.

*There are many salespersons who are prioritising targets [. . .] they must sell as much medicine as possible. At my place, if a salesperson comes, I'll just accept them and have conversation, but after that, buy medicine or not, most likely I don't buy it. (FARMER_02)*

Farmers also relied on other farmers in their network or farmer associations for information on disease management. One layer farmer who owned a poultry shop formed an informal collaboration with other farmers, providing them with disease management support and inputs through their technical services staff. Farmer associations also provided medication programs for independent farmers, much in the way integrators provide support to contracted farms.

**Lack of engagement with government services.** The interviews revealed that farmers preferentially engaged with non-government actors due to a lack of trust in the government's capacity to manage disease outbreaks. The government's management of the avian influenza crises was identified as contributing to farmers' reluctance to engage with government, with one interviewee noting that this necessitated farmers self-organising in networks or associations to address their problems.

*After all, the government is often not ready, especially when facing poultry disease outbreaks. [. . .] If we weren't prepared, we would have collapsed a long time ago [. . .]. For example, when there was a case of avian influenza in Indonesia around 2006 [. . .] the government was not ready to deal with the epidemic. We, layer farmers, have to look for vaccines independently. (ASSOCIATION_04)*

Farmers' trust in local government veterinary services was further undermined by a sense that the central government lacked understanding of the poultry sector's needs and that the government's priorities were not aligned with their own. For example, some interviewees highlighted the government's increased focus on highly pathogenic avian influenza at the

expense of diseases of greater economic importance to farmers. Farmers also expressed frustration regarding government control of input costs and egg and meat prices, feeling that their inability to change output prices to compensate for rising input costs had left them powerless.

> *Even though we are food heroes, we produce one of nine important staples. But there is no protection for us. Now, when the price of feed is expensive like last time, we have to get it from our own pockets. This isn't right. The government wants cheap, but from our money. (FARMER_06)*

This resulted in a reluctance for farmers to consult local government veterinary services for disease management, instead turning to contacts they trusted based on having worked successfully with them in the past.

> *Yes, we just never communicate with them [local government]. Except when it comes to licensing issues [...] But when it comes to disease, we often go directly to the veterinarian and several sources we can trust. (FARMER_08)*

In contrast, the technical services staff interviewed were aware of the importance of establishing trust with farmers for their advice to be taken seriously.

> *Surely they need time to trust us. [...] Only if they already trust us, otherwise they will inform us of nothing. (OTHER_11)*

Another reason for farmers' preference for technical services was that they were perceived as providing better support to farmers than the government. Interviewees told us that technical services staff responded faster, provided services such as diagnostic sample testing that the government did not, or had better access to information, which meant they were more likely to be contacted by farmers and had a closer relationship with them.

> *In broiler farmers and the farmers who get serviced from technical service [...] they will respond to that in at least in days, but in layer, you will need to wait for the officer from the government [...] probably they will report also to private technical service than to government, because usually they respond much faster. (OTHER_05)*

## Theme 3: Farmers' motivations and capacity to change AMU behaviour

Many of the interviewees had already begun to change the way in which they used antibiotics in recent years. The interviews highlighted farmers' main motivations to change AMU.

**Maintaining profitability in the context of fluctuating input price and quality.** Almost all interviewees reported that the main challenge for farmers was the increasing cost of feed and DOCs, which together accounted for 70–80% of farmers' input costs. This financial pressure was particularly felt by independent layer farmers who bore the burden of increased costs themselves, while contracted farms had fixed input prices. An inability to increase output prices led farmers to minimise the amount they spent on inputs, with antibiotics one of the only inputs that could be reduced.

> *The medicine is the money; we try not to use it. (MANAGER_01)*

*While antibiotics are quite expensive and the price of chicken feed is also high, antibiotics will only add to the cost. (ASSOCIATION_04)*

Farmers also responded to high input prices by changing to cheaper (and consequently lower quality) feed. This appeared to be a particular anxiety for farmers, who felt that this change had a significant impact on their production and disease control. Investment in good quality feed was seen as a way of reducing disease and subsequent AMU.

*If a sick chicken is only given medicine, but not given healthy food, it won't heal either. So one of the ways to minimise treatment is we are trying to make feed that is in accordance with the standards. (FARMER_11)*

Farmers were also challenged by fluctuations in input quality. Even when feed was affordable, farmers could not verify its quality, and told us that antibiotics were needed to compensate for the variable quality of feed.

*Yes, there are some manufacturers whose feed quality is really bad. But I handle that by giving the chickens prebiotics, enzymes and amino acid, sometimes antibiotics. (FARMER_02)*

**Adapting to regulations.** Decisions on AMU were made within the broader context of government policies to reduce AMU–in particular, a ban on antibiotic growth promoters (AGPs) legislated by the Indonesian government in 2018. Most interviewees mentioned the AGP ban, however, none raised the Indonesian government's more recent (2019) progressive ban on the use of colistin in livestock. A selection of interviewees' views on the impact of the AGP ban are summarised in S1 and S2 Tables.

Interviewees' opinions on the impact of the AGP ban differed. Although most interviewees told us they had themselves stopped using AGPs, many told us AGPs continued to be used by others. Some interviewees associated compliance with the ban with a drop in productivity and profitability, generating further financial pressure which was not necessarily offset by a concomitant decrease in expenditure on antibiotics. Even where interviewees were motivated to comply with the ban, farmers expressed frustration at the lack of government support for farmers and industry to make the transition.

*So, the AGP ban, it's okay, but we don't have the alternative. However, the expectation from the government is still the same. To produce low price protein products that are affordable. Actually [. . .] we all want to support this. [. . .] However, we're facing also the increase in raw material prices. (OTHER_04)*

There were exceptions, with some interviewees telling us the ban had no impact on or even caused an improvement in their production. However, one of the farmers who saw no impact had ceased to use AGPs years before the ban (FARMER_07), and it is possible that others who did not feel an impact were less reliant on their use, such that a ban would have only a marginal effect. Some interviewees were happy with the overall aims of the ban, noting that it was likely to lead to less AMR.

*Yes, performance has decreased [. . .] However, in the long term, this regulation might be better. Thus, the bacteria will be more sensitive to antibiotics. (OTHER_10)*

Most layer farmers responded to the ban by seeking replacements for AGPs, making use of feed additives including herbs, vitamins, amino acids, probiotics, enzymes, acidifiers, antitoxins, and formic acids, and reserving antibiotics for severe disease.

> *In principle for drugs, especially for antibiotics, we really try to minimize it. What we have on a regular basis are vitamins and additives such as amino acids and so on, that we use regularly. So we rarely prepare for antibiotics. We only use antibiotics if the chicken is really sick. (FARMER_06)*

> *As long as I use the herbs, I think I almost don't use antibiotics at all. (FARMER_05)*

**Feed additive use was less often mentioned by interviewees managing broiler farms.** Despite being widely used among the layer farmers interviewed, opinions on the efficacy and affordability of these products varied widely (presented in S3 Table). The purpose for which the feed additives were considered effective, or how farmers measured efficacy, was not always clear. However, even interviewees who thought feed additives were more expensive and less effective than antibiotics still used them, suggesting that they were seen as a necessity if antibiotics could no longer be used for growth promotion.

**Maintaining efficacy of drugs.**   The need to maintain drug efficacy was another factor driving farmers and technical services to change AMU practices. Some interviewees told us that they thought the antibiotics they used no longer worked as well as in the past, which they attributed to resistance.

> *In one city, amoxicillin can be very effective. But when I was in [another city], amoxicillin was not very effective. Because usually if the farmer is not directed by a veterinarian, he will continue to use one type of antibiotic and there is a possibility of resistance. (OTHER_14)*

Most farmers' concerns related to the potential impact of loss of drug efficacy on farm production. However, two farmers and an association representative also expressed concern about AMR as a public health problem and were worried about how their own AMU might be implicated.

> *I'm just sharing this, it's okay, in my mind, for example, this chicken has been using amoxicillin all its life, let's say it has amoxicillin residue, so it's consumed by humans and humans become resistant to amoxicillin. (FARMER_02)*

Farmers changed the way they used antibiotics to try to preserve the use of the drugs available to them. Some interviewees made decisions based on past AMU records. More rarely, antimicrobial sensitivity testing (AST) was performed. The manager of a large farm told us that they used AST routinely and modified treatments accordingly, though they did not specify where the testing was performed, or if the cost of testing was covered by the company to which they were contracted.

> *We would see [. . .] the historical data of antibiotics intake or usage [. . .] and in our farms internally every three periods we have a standard procedure of antibiotic resistance test. (MANAGER_02)*

An independent farmer (FARMER_04) put AST costs in perspective with overall treatment costs, suggesting that it was more cost effective to perform AST and target treatment

accordingly. Other farmers who did not have AST available to them instead used a rotating system, whereby they routinely changed the antibiotics used, although the rationale for antibiotic choice was not made clear in the interviews.

*Then, for the selection of antibiotics, I roll it, I don't have to use the same antibiotic all the time [. . .] basically I'm rolling, which one looks like it fits and is good for chicken. (FARMER_02)*

**Responding to consumer demand.**   Consumer demand was reported as a key influence for the interviewed farmers. However, most interviewees stressed that individual consumers were not interested in purchasing antibiotic-free poultry products, although larger supermarkets or restaurant chains might have requirements related to antibiotic residues.

*So we finally understand especially among the larger actors overuse of antibiotics would actually damage consumers, but we have to admit that actually for consumers themselves, they don't really have a lot of concern about this [. . .] concerns are only about pricing. (ASSOCIATION_02)*

Once again, cost was an overriding concern, with low prices outweighing considerations of other potential benefits that consumers might perceive. This was corroborated by the experience of a technical services officer who had assisted farmers to produce antibiotic residue–free and *Salmonella*-free eggs, but found they were required to sell these at a loss due to lack of consumer interest.

*Our team is trying to make eggs free of antibiotic residues and we check to the lab, until it is free of antibiotics, free of salmonella [. . .] but there is no market here. (OTHER_14)*

**Maintaining and improving high farming standards and practices.**   The farmers and technical services interviewed were also motivated to reduce AMU out of a more generalised interest in improving their management practices. Reducing AMU appeared to be both a part of and a necessary consequence of improved farm management, such that routine AMU was considered an indication of poor farm management.

*If farmers manage the farm well, they rarely use antibiotics because their chickens rarely get sick. (ASSOCIATION_04)*

*If they want to reduce the antibiotic they have to provide the good management [. . .] they have to change from open house to close house. (OTHER_09)*

Good management was usually considered to consist of closed housing and application of other unspecified biosecurity measures. The increased influence of integrators appeared to be accelerating the existing trend towards intensive poultry farming systems in Indonesia for contracted farmers, as some companies refused to enter into new contracts with farms that did not have closed housing. For independent farmers who usually maintained semi-open or open housing systems, biosecurity measures were nonetheless considered important to ensure good farm management, although the specific measures farmers understood to be part of farm biosecurity were not clearly stated. Biosecurity measures were considered a worthwhile financial investment if their adoption could be offset by reduced expenditure on antibiotics.

*With biosecurity, the operational costs will increase [. . .] but it's paid for, as the use of antibiotics is down by 70 percent. (FARMER_07)*

Use of other measures to reduce or change AMU, such as AST to direct antibiotic choice, were considered by some farmers as part of a farmer's professional responsibility.

**Interviewer:** Did you follow the results of the sensitivity test when treating chickens?

***Interviewee:*** *That's right. So, we would not try various drugs in vain and so as not continue to blame the drug manufacturers. It is a bad attitude. (FARMER_04)*

Similarly, technical services staff emphasised the importance of providing advice on farm management rather than prioritising sales, for the sake of their professionalism and reputation. This differed from the perceptions of FARMER_02 (mentioned above) that technical services privileged sales above all else.

*I'm a bit different from the others, I'm more inclined to investigate what's going on [. . .]. For example, if the broiler is about to harvest, we don't recommend using antibiotics [. . .] because I myself am more concerned by the importance of my reputation. (OTHER_14)*

## Discussion

This study identified many factors that interviewees believed influenced their AMU decisions. These factors can be conceptualised as barriers or facilitators to changing AMU behaviour, to understand how they could be used to better improve development of AMU-related interventions.

### Barriers to improving AMU practices

The patterns of AMU reported by farmers and technical services in this study suggested that prophylactic use was common and not well distinguished from treatment, as has been reported elsewhere in Southeast Asia [5, 44]. In many cases, this seemed to be due to both limited access to resources such as diagnostic testing, and a lack of shared understanding of when antibiotics were necessary. The farmers interviewed told us they used antibiotics only when disease was severe. However, further probing indicated that farmers' perception of severity varied, with acceptable levels of mortality or morbidity ranging from 1% to 30%. This suggests a need for better access to diagnostic tools and more appropriate guidance on when to use antibiotics tailored for local conditions.

Another major constraint was the impact of external financial pressures which restricted farm management and left farmers feeling they had limited opportunity to improve their AMU or disease management practices. Farmers tended to respond to financial pressure by reducing expenditure on inputs, notably antibiotics. This finding was unexpected, as past research has indicated that medicines comprise only a small proportion of farm input costs in Southeast Asia, and are unlikely to influence AMU decisions [45, 46]. However, in the broader context of rising input costs across Indonesia, it appeared that antibiotics were one of the only inputs farmers could reduce, suggesting that the importance of antibiotic cost is contextually dependent. While reduced input expenditure could result in reduced AMU due to fewer antibiotic purchases, such behaviour could also lead to increased AMU, due to purchases of poorer quality inputs and reduced investment in biosecurity. There was a general consensus among

interviewees that reduced input quality meant that some degree of routine (i.e. prophylactic) AMU was inevitable, which mirrors findings in Europe that veterinarians thought prophylactic AMU was inevitable due to poor quality feed or DOCs [47]. Given that many farmers and veterinarians' intentions and behaviours to reduce AMU are associated with their belief in the feasibility of AMU reduction [48], the findings from our study suggest that achieving positive change in AMU behaviours in Indonesia is likely to be challenging without support to address what farmers consider to be insurmountable barriers.

Finally, many interviewees expressed a lack of trust in the government, which made them reluctant to report disease to or request assistance from local veterinary services. This has likely hampered government efforts to manage diseases and promote responsible AMU. However, according to the interviewees, the government's ban on AGPs in livestock [49] had a positive impact in changing AMU. This suggests that government regulation can influence behaviour, consistent with findings of the effectiveness of different kinds of regulation in reducing farmers' AMU [50]. Future research could identify the factors that facilitate success of such regulation, although it is likely that addressing systemic factors will play a major role [19].

## Opportunities for changes in AMU practices

Several opportunities to address changes in AMU-related behaviours can be drawn from this study. In contrast to studies describing low levels of awareness among farmers of AMU/AMR and associated government policies [6, 24, 29, 51], we found that interviewees were well-aware of antibiotics as a distinct class of medicines as well as government policy regarding AMU, and were motivated to modify their AMU. Notably, two layer farmers raised the potential risks for human health of AMR. These findings suggest an opportunity in Indonesia for engaging with farmers with high awareness and motivation, to pilot programs in which they can act as champions, or to generate more targeted training programs on how to reduce AMU within the constraints that they face. Awareness campaigns targeting the broader public may also have an impact, as interviewees told us that if consumer demand shifted to a preference for antibiotic-free meat, farmers would modify their behaviour accordingly. However, while knowledge and awareness of AMU/AMR can be important contributors to behavioural change, they are unlikely to effect change in the absence of addressing more systemic barriers to changing AMU [19, 28, 52]. The interviews indicated that even farmers who were aware of the potential benefits of reducing AMU found it difficult to do so due to economic pressures and a lack of viable alternatives. This likely means that it will be important to facilitate investment in infrastructure for improved biosecurity and provision of viable alternatives to antibiotics alongside efforts to capitalise on existing awareness.

Another potential opportunity to better engage with farmers is through leveraging the influence of technical services staff and farmers' associations. Veterinarians are often considered the most trusted source of advice and technical support by farmers [53], as are non-government animal health workers and veterinary paraprofessionals in countries where access to government veterinarians are limited [54]. However, in our study we found that it was rather technical staff employed by the private sector and peer networks who were preferentially trusted by farmers when seeking advice. Farm consultants from pharmaceutical companies have also been identified as important sources of AMU advice in Thailand [27], suggesting that the private sector may be more important across the Southeast Asian region. Delivering antimicrobial stewardship messages through these actors may be more effective than through others, as farmers may be more likely to follow advice provided by a trusted source. Additionally, participatory approaches based around social engagement with peer networks have been

shown to effect a change in AMU in Europe [55, 56], and the finding that farmers rely on peers and support staff suggests that such approaches may also meet with success in Indonesia.

Lastly, farmers and technical services staff thought that reducing AMU by investing in biosecurity and antibiotic alternatives were practices associated with being a good farmer. This echoes findings in Europe that changes in AMU can come as part of broader engagement in good farming practices [57], and in Southeast Asia that some farmers believe well-run farms should not need antibiotics [27]. A sense of identity has been seen to be important for promoting sustainable behavioural change among farmers [20], and suggests that attempts to frame AMU reduction within farmers' conceptions of good farming may be useful for developing future programs. However, there were considerable differences of opinion between the interviewees as to what good biosecurity practices were, as well as on the effectiveness of feed additives. Presenting investment in biosecurity and other measures as a means of reducing expenditure on AMU, along with clearer information on what impact these approaches can have, may be one point of leverage for changing behaviour.

## Study limitations and considerations

The deliberate inclusion of interviewees who were motivated and interested in discussing their AMU facilitated an in-depth discussion of the factors influencing AMU practices from their perspective. However, our results should be interpreted within this context and should not be taken as representative of the broader groups from which the interviewees came. The findings are not intended to be representative of the Indonesian poultry sector as a whole, but rather to provide an in-depth analysis of the views of a specific set of actors in the poultry sector, which could help guide directions or generate hypotheses for future qualitative or quantitative investigations of this topic. This is aligned with common practice in qualitative research, particularly qualitative research undertaken with an interpretivist framework, in which depth of analysis is prioritised [40].

The interviews took place both online and in-person. Although the use of online tools could be seen as a potential limitation, this is unlikely to have had an impact on the analysed content of the interviews–research investigating the impact of different modalities of data collection in qualitative research have found that while using online interviews may slightly reduce the quantity of data compared to in-person interviews (as measured by number of words and statements), it does not impact the thematic content of responses [58, 59].

The predominance of layer farmers among the interviewees likely means that the views of contracted farmers (who are predominantly broiler farmers) were underrepresented. However, the deliberate focus on the layer sector specifically addressed a gap in the literature, as much of the past research on poultry farmers' AMU practices in Southeast Asia has focused on AMU in broilers, or both broilers and layers together [5, 7, 11, 60, 61]. Our findings on AMU practices among layer famers were similar to those of the few broiler farm managers we interviewed, and what has been reported in the literature previously for broiler farmers [11]. In particular, our results showed that prophylactic AMU was common among the independent layer farmers interviewed, and that AMU was particularly important at DOC arrival and pre- and post-vaccination. We also found that layer and broiler farmers had similar sources of technical advice, with independent layer farmers also engaging extensively with technical services from private sector companies, despite not having formal contracts with these actors. This was surprising, as broiler and layer farms in Indonesia have different structures and business models, which might have been expected to manifest in a different approach to AMU.

To build on the findings from this study, it would be important to hear perspectives from other stakeholder groups not consulted in this study–most notably the government.

Preliminary discussions were held with some government officials, but the content of these discussions was not usable for this study, and additional meetings could not be arranged. It would be important for future research on AMU practices to target government officials for inclusion in order to better understand the barriers farmers told us existed to engaging with government veterinary services, and the government's perspective on these barriers. Future studies could also target inclusion of female farmers–all but two of the farm workers and managers we spoke to were male, and it has been suggested that gender may play a role in influencing AMU practices in Southeast Asia [62].

During the interviews, we did not hear from any interviewees regarding accessibility or the quality of antibiotics, which was unusual given that the quality of other inputs was of paramount importance to them. Limited access to good quality antibiotics has been shown to be an important challenge to responsible AMU in other LMICs [63]. Several interviewees mentioned a lack of antibiotic efficacy which they attributed to resistance, although this could also have been due to poor antibiotic quality. Future work should incorporate considerations of antibiotic access and quality to better understand the role they may have in influencing AMU decisions.

The complex interplay of different factors identified in this study that influence AMU decisions makes it difficult to draw conclusions about how farmers' AMU decisions may be impacted by a change in any one of these factors (for example, were input prices to decrease, it is not clear that this would result in a subsequent increase in AMU) or by an external intervention. Further analysis would be required to evaluate this. Quantitative approaches such as factorial survey analysis have been used to investigate how factors identified in social theories impact veterinarians' prescribing behaviours in different scenarios [64]–there is the potential to use a similar approach to assess the relative importance of factors that we identified as influencing farmers' AMU.

The interpretivist framework used in this study facilitated analysis of a complex issue (AMU practices) from multiple stakeholder perspectives, however, this approach necessitates additional consideration and self-reflection of how researcher subjectivity influences the interpretation and credibility of the results [34, 65]. Several approaches were made to take this into consideration. During the interviews, short summaries were frequently repeated back to the interviewee by the person conducting the interview to determine if the notes reflected their experiences as they had described them and to identify where interpretations differed. During thematic development, extensive self-reflection was performed to ensure that themes developed were grounded in the data, and to avoid placing undue emphasis on the transcripts which the first author had transcribed and was therefore most familiar with. In the results, in-depth and contextualised descriptions were provided alongside examples of participants' statements to illustrate how they related to the identified themes.

## Conclusion and recommendations

These findings have broader implications for how to facilitate behavioural change in AMU practices. Identification of the barriers which limit different actors' abilities to reduce AMU will better allow these to be considered during the planning and implementation of interventions. The results also suggest that farmers' existing awareness of AMU should be leveraged with awareness campaigns tailored to their identified needs, and to those of consumers. Communication of messages related to antimicrobial stewardship could be delivered through better engagement with the private sector and farmers' associations in addition to existing interlocuters, as well as via approaches based around social engagement within peer networks. Lastly,

AMU interventions may be improved by engaging with farmers' conceptions of what it means to be a good farmer.

## Supporting information

**S1 Appendix. Standards for Reporting Qualitative Research (SRQR) checklist.**
(PDF)

**S2 Appendix. Example of interview guide.**
(PDF)

**S1 Table. Interviewees' opinions on the impact on production and profitability of the 2018 ban on antibiotic growth promoters in livestock.**
(PDF)

**S2 Table. Interviewees' opinions on the impact on AMU and AMR of the 2018 ban on antibiotic growth promoters in livestock.**
(PDF)

**S3 Table. Summary of interviewees' opinions on the price and efficacy of feed additives as compared to antibiotic growth promoters.**
(PDF)

## Acknowledgments

The authors would like to thank the participants who took part in this study, and Dr. Skye Badger for inputs on the final draft of the manuscript.

## Author Contributions

**Conceptualization:** Rebecca Hibbard, Lorraine Chapot, Havan Yusuf, Kurnia Bagus Ariyanto, Kusnul Yuli Maulana, Angus Cameron, Timothée Vergne, Céline Faverjon, Mathilde C. Paul.

**Data curation:** Rebecca Hibbard, Lorraine Chapot, Havan Yusuf, Kurnia Bagus Ariyanto, Kusnul Yuli Maulana, Widya Febriyani.

**Formal analysis:** Rebecca Hibbard.

**Investigation:** Rebecca Hibbard, Lorraine Chapot, Havan Yusuf, Kurnia Bagus Ariyanto, Kusnul Yuli Maulana, Widya Febriyani, Angus Cameron.

**Methodology:** Rebecca Hibbard, Lorraine Chapot, Havan Yusuf, Kurnia Bagus Ariyanto, Kusnul Yuli Maulana, Angus Cameron, Timothée Vergne, Céline Faverjon, Mathilde C. Paul.

**Supervision:** Céline Faverjon, Mathilde C. Paul.

**Validation:** Rebecca Hibbard, Lorraine Chapot, Havan Yusuf, Kurnia Bagus Ariyanto, Kusnul Yuli Maulana, Widya Febriyani, Angus Cameron, Timothée Vergne, Céline Faverjon, Mathilde C. Paul.

**Writing – original draft:** Rebecca Hibbard, Céline Faverjon, Mathilde C. Paul.

**Writing – review & editing:** Rebecca Hibbard, Lorraine Chapot, Havan Yusuf, Kurnia Bagus Ariyanto, Kusnul Yuli Maulana, Widya Febriyani, Angus Cameron, Timothée Vergne, Céline Faverjon, Mathilde C. Paul.

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
