## [Decision Letter · Decision Letter 0]

26 Jun 2023

PONE-D-23-16108

“It's a habit. They've been doing it for decades and they feel good and safe.”: A qualitative study of barriers and opportunities to changing antimicrobial use in the Indonesian poultry sector

PLOS ONE

Dear Dr. Hibbard, 

Thank you for submitting your manuscript to PLOS ONE. After careful consideration, we feel that it has merit but does not fully meet PLOS ONE’s publication criteria as it currently stands. Therefore, we invite you to submit a revised version of the manuscript that addresses the points raised during the review process.

We look forward to receiving your revised manuscript.

Kind regards,

Sk Shaheen Islam, DVM, MS, Ph.D.

Academic Editor

PLOS ONE

Journal Requirements:

Additional Editor Comments:

The authors' utilization of the Standards for Reporting Qualitative Research (SRQR), which includes 21 items, is not adequately addressed in the current version of the manuscript. For example, the unit of study is not clearly defined and explained by the authors. It is crucial for the authors to address this issue in the revised manuscript and provide a comprehensive explanation of each item.

There have been studies and reports highlighting the prevalence and impact of AMR and AMU in Indonesia. The authors should consider discussing these findings in the introduction section to provide a comprehensive insights of the situation and this absent in this manuscript.

Reviewers' comments:

Reviewer's Responses to Questions

**Comments to the Author**

1. Is the manuscript technically sound, and do the data support the conclusions?

Reviewer #1: Yes

Reviewer #2: Yes

Reviewer #3: Partly

2. Has the statistical analysis been performed appropriately and rigorously? 

Reviewer #1: I Don't Know

Reviewer #2: N/A

Reviewer #3: No

3. Have the authors made all data underlying the findings in their manuscript fully available?

Reviewer #1: No

Reviewer #2: Yes

Reviewer #3: No

4. Is the manuscript presented in an intelligible fashion and written in standard English?

Reviewer #1: Yes

Reviewer #2: Yes

Reviewer #3: No

5. Review Comments to the Author

Reviewer #1: It’s a qualitative study designed to focus different barriers and opportunities to changing antimicrobial use in the Indonesian poultry sector.

Major comment:

This is a nice study, however, the major concern in the sample size, e.g., only 35 interviews. The poultry sector is very big in Indonesia. There are so many farms. The authors must have to define that these 35 interview is enough to claim the findings or conclusions of this study is accurate/ genuine/true reflection of the poultry sector??

Other comments:

Lone 88, what was the basis of selecting that wide range of stakeholders?

Line 122, how od you know that a sufficient diversity of views was covered?

Table 2, who were those “other” groups?

What was the major difference in the responses/finding between layer farmers and broiler farmers??.. discuss briefly in a short para separately..

Line 548, separate “study limitation” from “implication”, and write “Conclusion and recommendation” instead of implications as a new subheading.

Reviewer #2: General Comments:

The authors did a great job of qualitative research on barriers and opportunities to changing antimicrobial use in the Indonesian poultry sector. However, the number of interviewees is only 35 from different sectors. The Government representative should be included. The roles of farm owners, managers and workers and their views of AMU in farms are different. Are all interviewees faced with all questions or opinions? If so, Table S1, S2 and S3 dictated only a few participants to give their views.

Specific comments:

Lines 46-47: As it is different paragraph, complete the sentence……Interventions that promote responsible and prudent use of antimicrobials have the potential to mitigate the risk of AMR.

Line 122-23: How did you determine initial sample size? What was the criteria to be sufficient diversity of views?

Line 197: Is FARMER_06 the same interviewee as ASSOCIATION_04?

Line 201: Almost all interviewees reported routinely to whom or where?

Line 223: How about other interviewees for production parameters used to inform antibiotic use decisions and the associated actions taken by them? They didn’t provide any statement regarding this issue?

Line 285: As per the statement of interviewees on “Lack of engagement with government services”, why this study didn’t consider any government officials to include?

Line 330-331: When feed and DOCs cost will be lower, farmers will use more antibiotics like previous time. Are these types of questions asked to interviewees?

Line 352: Is there any antibiotics that is prohibited to use in poultry in Indonesia?

Line 407: Where did the large farm perform AST?

Line 548: Conclusion should be separate

Reviewer #3: 1. The methodology of the research work is not clear and well justified

2. Stakeholder interviews were taken in-person and online. Online interviews of the farmers are not well acceptable

3. There are no demographic findings of the interviewees though it was essential

4. No statistical tool was used to find out correlation with the findings

5. There are no measurable and substantial findings of the research work

6. PLOS authors have the option to publish the peer review history of their article (what does this mean?). If published, this will include your full peer review and any attached files.

Reviewer #1: **Yes: **Md. Tanvir Rahman

Reviewer #2: No

Reviewer #3: No

---

## [Author Response · Author response to Decision Letter 0]

25 Jul 2023

Responses to editor’s comments:

Editor: The authors’ utilization of the Standards for Reporting Qualitative Research (SRQR), which includes 21 items, is not adequately addressed in the current version of the manuscript. For example, the unit of study is not clearly defined and explained by the authors. It is crucial for the authors to address this issue in the revised manuscript and provide a comprehensive explanation of each item.

The completed Standards for Reporting Qualitative Research (SRQR) checklist has been attached, with line numbers indicating where each item has been addressed. For transparency, this has now been added to the supplementary information (S2 Appendix). It is referred to in the manuscript in lines 93-94 and 830.

Editor: There have been studies and reports highlighting the prevalence and impact of AMR and AMU in Indonesia. The authors should consider discussing these findings in the introduction section to provide a comprehensive insights of the situation and this absent in this manuscript.

This has been addressed in the manuscript (lines 46-57). The additional text reads:

“In Indonesia, high levels of AMU have been detected on broiler farms, with frequent overdosing and underdosing relative to recommended dosages, and high levels of prophylactic AMU alongside lower use for growth promotion, although the boundaries used by farmers to distinguish between AMU for treatment, prophylaxis, or growth promotion are often unclear [1,2]. The presence of disease, farmer aspiration to prevent disease, and desire for improvement in productivity and growth have been identified as motivators for AMU among Indonesian broiler farmers, with contract farmers more likely than independent farmers to use AMs to prevent disease. [3]. However, most research to date has concentrated on broiler farmers contracted to large integrated companies, with less data available on layer farmers (who tend to be independent farmers in Indonesia). In light of reports of high levels of multi-drug resistant bacteria in bacterial isolates from live poultry and chicken meat throughout Indonesia [4,5], efforts to address AMR are of increasing importance in the country and the Southeast Asia region more broadly.”

 

Responses to reviewers’ comments:

Responses to comments about sampling

Response to the following comments on sample size:

- Reviewer 1 (general comments): This is a nice study, however, the major concern in the sample size, e.g., only 35 interviews. The poultry sector is very big in Indonesia. There are so many farms. The authors must have to define that these 35 interview is enough to claim the findings or conclusions of this study is accurate/ genuine/true reflection of the poultry sector??

- Reviewer 2 (general comments): The authors did a great job of qualitative research on barriers and opportunities to changing antimicrobial use in the Indonesian poultry sector. However, the number of interviewees is only 35 from different sectors. 

We note that a number of comments from reviewers 1 and 2 related to sample size – we have provided what we feel is a comprehensive response below. Modifications have also been made to the manuscript in the methodology in lines 136-149 and the results in lines 575-580 summarising the longer explanation below.

This is a qualitative research paper, in which the purpose was to explore the factors influencing AMU practices in the Indonesian poultry sector. We are not claiming that the findings are representative of the Indonesian poultry sector as a whole, but rather that they provide an in-depth analysis of the views of a specific set of actors in the poultry sector, which could help guide directions or generate hypotheses for future qualitative or quantitative investigations of this topic. This is aligned with common practice in qualitative research, particularly qualitative research undertaken with an interpretivist framework, in which depth of analysis is prioritised. For this reason, sample sizes in qualitative research tend to be much smaller than those used in quantitative research [6]. It should also be noted that qualitative research tends to use purposive sampling (as compared to the probabilistic sampling used in quantitative research), in order to facilitate inclusion of participants who can provide “information-rich” data relative to the research question [7].

We would argue that a sample size of 35 interviews is relatively large compared to what is recommended in the literature on qualitative research [6–9]. We note this not to suggest that a larger number is better – the appropriate sample size for a study will depend on the study design, methodology, and research aim, among other factors – but rather to qualify the remarks below and in the manuscript where we explain the rationale for the sample size, and refer to our sample size as “large”. While we determined the necessary sample size based on the criteria outlined below, rather than external guidance, we note also that the sample size of our research (35 interviews) is nonetheless in line with the approximate sample size suggested for studies using semi-structured interviews (30-60 interviews) and grounded theory (20-30 or 30-50 interviews,) [7,10].

The sample size was determined both pragmatically and with regards to the concept of information power, whereby sample size is considered within the context of the following criteria: study aim, sample specificity, established theory, quality of dialogue, and analysis strategy [11]. A relatively large sample size was used based on the needs identified by these criteria – the broad, exploratory nature of the study aim and analysis strategy, the sample specificity (the target to include multiple stakeholders from several different groups of stakeholders), and the highly variable quality of dialogue identified during data collection.

Response to the following comment on stakeholder identification:

- Reviewer 1: Line 88, what was the basis of selecting that wide range of stakeholders?

The range of stakeholders to be included was based on a stakeholder analysis of important actors in the Indonesian poultry sector that was performed in advance of the study (mentioned in lines 125-126.) “Participants were identified and prioritised through a stakeholder analysis of important actors in the Indonesian poultry sector”). Line 88 (now lines 100-102) has been revised to reflect this also. It now reads: “A wide range of different stakeholders were consulted to ensure a diversity of views, identified based on a stakeholder analysis of the Indonesian poultry sector.”

Response to the following comments on determining sample size based on diversity of views:

- Reviewer 1: Line 122, how do you know that a sufficient diversity of views was covered?

- Reviewer 2: Line 122-23: How did you determine initial sample size? What was the criteria to be sufficient diversity of views?

Initial sample size was determined based on the number of different stakeholder groups we wanted to consult, assuming that we would include multiple interviewees within each category, placing a stronger priority on farmers as it was particularly their practices which we were interested in. We also made an approximate estimate of the number of interviews likely needed to ensure a sufficient depth of analysis. The initial sample size was also based pragmatically on the number of interviewees we expected that we could reasonably meet with in the timeframe allocated for the study. 

Deciding when a sufficient diversity of views was met was a judgement that we made during the data collection – we stopped adding new participants when we felt that a sufficient diversity of views, in interviews of sufficient depth, had been collected for us to be able to address our research question, while taking into account the criteria relating to information power (see above). There are many different approaches which can be used in qualitative research to determine when to stop adding new interviewees, such as saturation, pragmatic considerations, quality of analyses, externally imposed sampling requirements, or sampling guidelines [6]. We used considerations of analytic quality and pragmatism, stopping at the point where we considered that the data collected was sufficiently rich to allow identification and a pattern-based analysis of the factors influencing AMU and their importance to our interviewees. We fully acknowledge that this is a value judgement – more information can always be collected, but the question is rather whether this permits an analysis of sufficient depth, and we considered that it was so. 

Additional text (lines 143-149) has been added to elaborate on these points in the manuscript: 

“An initial sample size of 20 was targeted based on the number of different stakeholder categories identified in the stakeholder analysis, assuming that multiple interviewees would be included within each category, and considering the number of interviewees that could reasonably be met with in the timeframe allocated for the study. New participants were gradually included until it was judged that a sufficient diversity of views was collected – when it was considered that the data collected was sufficiently rich to allow identification and analysis of the factors influencing AMU.”

Response to the following comments on inclusion of government representatives:

- Reviewer 2 (general comments): The Government representative should be included. 

- Reviewer 2: Line 285: As per the statement of interviewees on “Lack of engagement with government services”, why this study didn’t consider any government officials to include?

This has been addressed in the manuscript (lines 600-606):

“To build on the findings from this study, it would be important to hear perspectives from other stakeholder groups not consulted in this study – most notably the government. Preliminary discussions were held with some government officials, but the content of these discussions was not usable for this study, and additional meetings could not be arranged. It would be important for future research on AMU practices to target government officials for inclusion in order to better understand the barriers farmers told us existed to engaging with government veterinary services, and the government’s perspective on these barriers.“

Responses to all other comments on content

Reviewer 1: Table 2, who were those “other” groups?

“Other” in Table 2 is part of the anonymised ID that were assigned to certain interviewees (all those who were not who were not associations, farmers, or farm managers). This list includes pharmaceutical companies, universities, feed companies, international organisations, integrators, poultry shops, and research institutions.

The specific category and role of each interviewee (including those which have been labelled with ”Other”) are provided in Table 1 (page 10). A legend has been added to Table 2 to refer readers back to Table 1 for this information (lines 246) which reads “Please refer to Table 1 for information on the interviewees referred to by their ID in this table.”

Reviewer 1: What was the major difference in the responses/finding between layer farmers and broiler farmers? discuss briefly in a short para separately.

All farmers that we interviewed were layer and/or pullet farmers. The interviewees from broiler farms were mangers or supervisors, rather than farmers – nonetheless, we did not find major differences with regards to AMU practices between these two groups, or between layer farmers and information available on broiler farmers in the literature. The discussion has been expanded to include this (lines 587-599). This paragraph now reads:

“The predominance of layer farmers among the interviewees likely means that the views of contracted farmers (who are predominantly broiler farmers) were underrepresented. However, the deliberate focus on the layer sector specifically addressed a gap in the literature, as much of the past research on poultry farmers’ AMU practices in Southeast Asia has focused on AMU in broilers, or both broilers and layers together [3,12–15]. Our findings on AMU practices among layer famers were similar to those of the few broiler farm managers we interviewed, and what has been reported in the literature previously for broiler farmers [3]. In particular, our results showed that prophylactic AMU was common among the independent layer farmers interviewed, and that AMU was particularly important at DOC arrival and pre- and post-vaccination. We also found that layer and broiler farmers had similar sources of technical advice, with independent layer farmers also engaging extensively with technical services from private sector companies, despite not having formal contracts with these actors. This was surprising, as broiler and layer farms in Indonesia have different structures and business models, which might have been expected to manifest in a different approach to AMU.”

Response to the following comments on asking questions to all participants:

- Reviewer 2 (general comments): The roles of farm owners, managers and workers and their views of AMU in farms are different. Are all interviewees faced with all questions or opinions? If so, Table S1, S2 and S3 dictated only a few participants to give their views.

- Reviewer 2: Line 223: How about other interviewees for production parameters used to inform antibiotic use decisions and the associated actions taken by them? They didn’t provide any statement regarding this issue?

The semi-structured interview guide (an example of which is also provided in the supplementary findings, S1 Appendix) consisted of a list of topics and prompting questions. Participants were free to elaborate on specific topics from this list or even raise new subjects which were of most interest to them. Not all questions were addressed to all participants, to allow participants the freedom to discuss in greater depth the topics which were most relevant to their interests. This is consistent with approaches taken in other qualitative studies using semi-structured interview guides (see for some examples: [16–19]), and with the interpretive framework that we used which prioritised interviewee’s perspectives.

The supplementary tables S1, S2, and S3 provide selected extracts from the transcripts on the topics of the impact of the 2018 ban on antimicrobial growth promoters on production and profitability, on AMU and AMR, and the price and efficacy of feed additives, respectively. The excerpts in the supplementary tables are included to provide support for the conclusions drawn in the results and discussion, as an example of the material on which our findings were based. Not all participants are included in these tables, because not all participants expressed an opinion on each of these topics.

Reviewer 2: Line 197: Is FARMER_06 the same interviewee as ASSOCIATION_04?

Yes. The legend for Table 1 (now line 219) has been revised to read: “FARMER_06 is the same interviewee as ASSOCIATION_04, who was interviewed on two separate occasions, being interviewed in their capacity as a farm owner, and as the head of an association respectively.”

Reviewer 2: Line 201: Almost all interviewees reported routinely to whom or where?

The word “reported” is here being used in the sense of “stated”, or “informed”. The interviewees reported to us, the interviewers, the fact that they routinely used antibiotics or recommended their use – this was not meant to imply reporting antimicrobial use to an authority. This sentence has been changed for clarity (lines 223-225). It now reads “Almost all interviewees indicated during interviews that they routinely used antibiotics or recommended their use, with decisions based on the information directly available to them at the time.”

Reviewer 2: Line 330-331: When feed and DOCs cost will be lower, farmers will use more antibiotics like previous time. Are these types of questions asked to interviewees?

No. Questions regarding different specific scenarios were not asked to interviewees. It is difficult to assume that this would be the case without knowing what other changes might occur in other factors influencing AMU decisions. To address this point, additional text has been added to the discussion (lines 616-623):

“The complex interplay of different factors identified in this study that influence AMU decisions makes it difficult to draw conclusions about how farmers’ AMU decisions may be impacted by a change in any one of these factors (for example, were input prices to decrease, it is not clear that this would result in a subsequent increase in AMU) or by an external intervention. Further analysis would be required to evaluate this. Quantitative approaches such as factorial survey analysis have been used to investigate how factors identified in social theories impact veterinarians’ prescribing behaviours in different scenarios [20] – there is the potential to use a similar approach to assess the relative importance of factors that we identified as influencing farmers’ AMU.”

Reviewer 2: Line 352: Is there any antibiotics that is prohibited to use in poultry in Indonesia?

Yes, the use of colistin in livestock was banned in 2019 [21]. A mention of this has now been added to the manuscript (lines 374-377):

“Decisions on AMU were made within the broader context of government policies to reduce AMU – in particular, a ban on antibiotic growth promoters (AGPs) legislated by the Indonesian government in 2018. Most interviewees mentioned the AGP ban, however, none raised the Indonesian government’s more recent (2019) progressive ban on the use of colistin in livestock.”

Reviewer 2: Line 407: Where did the large farm perform AST?

We did not ask farmers to tell us the laboratories in which they performed antimicrobial sensitivity testing. However, as the interviewee from the large farm who told us this (MANAGER_02) was the manager of broiler farms in an integrated company, it is likely that testing was performed by a private laboratory owned by or affiliated with the integrator company. Alternative possibilities for AST would be regional government laboratories or university laboratories. We have not speculated on the possible sites of testing in the results section of the manuscript, but we have added additional text to clarify that this manager was associated with a contracted farm who may have provided the cost of the testing, but that this information was not provided (lines 428-431):

“More rarely, antimicrobial sensitivity testing (AST) was performed. The manager of a large farm told us that they used AST routinely and modified treatments accordingly, though they did not specify where the testing was performed, or if the cost of testing was covered by the company to which they were contracted.”

Reviewer 3: The methodology of the research work is not clear and well justified

We have reported the methodology of our research (the general approach and the analytic method), which follow well-established practices in qualitative research. 

We discuss the general approach to the methodology in the section “Qualitative approach and framework” (lines 97-111). Here we have indicated that our approach used elements of constructivist grounded theory as described in [22]. Additional text has now been added to the manuscript (lines 103-106) to add more detail on what these elements are, and this sentence now reads: “The analytic framework used for this study borrowed aspects of constructivist grounded theory as espoused by Charmaz [22] (in particular, incorporating the view that both the data and the resulting analyses are constructed from the experiences of the participants and researchers, and basing the results on the data itself).”

We discuss the analytic method used in the section “Data analysis, general approach” (lines 190-199). The manuscript reads (lines 192-194): “Reflexive thematic analysis following the method of Braun and Clarke [23,24] was employed, whereby the researcher plays an active role in engaging with and generating themes from the data to identify patterns of shared meaning.”

We believe the use of the approach and analytic method described above is justified by their suitability to address our research aim to explore the factors influencing AMU practices, from the viewpoints of a range of different stakeholders.

Reviewer 3: Stakeholder interviews were taken in-person and online. Online interviews of the farmers are not well acceptable

Many other studies using qualitative and quantitative research methods have made use of online interviews, which would suggest that this method is broadly acceptable to the scientific community. Furthermore, research investigating the impact of different modalities of data collection in qualitative research (in particular online vs in-person methods) have not found this to have a significant impact on results. We have now addressed this in the manuscript (lines 581-586):

“The interviews took place both online and in-person. Although the use of online tools could be seen as a potential limitation, this is unlikely to have had an impact on the analysed content of the interviews – research investigating the impact of different modalities of data collection in qualitative research have found that while using online interviews may slightly reduce the quantity of data compared to in-person interviews (as measured by number of words and statements), it does not impact the thematic content of responses [25,26].”

Reviewer 3: There are no demographic findings of the interviewees though it was essential

We have added some demographic information (the gender of the interviewees) to Table 1 (page 10). We note that other interviewee attributes of pertinence to the study results – farm size and type, and role of the interviewees – are already presented in Table 1. We have also added the following text to the discussion (lines 606-608):

“Future studies could also target inclusion of female farmers– all but two of the farm workers and managers we spoke to were male, and it has been suggested that gender may play a role in influencing AMU practices in Southeast Asia [27].”

Reviewer 3: No statistical tool was used to find out correlation with the findings

The necessity of using statistical approaches such as correlation should be determined according to a given study’s research aim and objectives. This is a qualitative research paper, in which the research aim and methodology did not necessitate use of a statistical tool. In general, statistical methods are more appropriate for use in quantitative research. Qualitative research tends to address research questions which are more exploratory in nature, and the method of data analysis applied is often inductive and used to establish patterns or themes [28].

In this study, we were not aiming to identify correlation between different factors – rather, the research aim of this paper was to explore the factors influencing AMU practices in the Indonesian poultry sector (lines 80-82). We used a qualitative approach to address this research aim, in order to provide a contextualised understanding of non-quantitative factors that influence and impact decision-making related to AMU, from the perspective of the farmers and other stakeholders that we interviewed.

To address our research aim, the analytic method used was reflexive thematic analysis, a pattern-based approach to data analysis in which the researcher reads and engages with the dataset (in our case, interview transcripts) in order to construct themes based on patterns of shared meaning [23,24], described in lines 192-194. The output or results from such an analysis is a description of the identified themes and what these can tell us about the dataset. It does not involve the use of correlation or other statistical methods.

We note also that our manuscript follows the recommended standards for reporting on qualitative research (SRQR) [29], now attached in S2 Appendix, which do not necessitate the use of statistical tools. Please refer to this Appendix which identifies the sections of the manuscript which meet these standards’ requirements for data analysis and results/findings.

Reviewer 3: There are no measurable and substantial findings of the research work

Qualitative research is unique for its ability to provide in-depth, contextualised analyses of individuals’ experiences and the meanings that they ascribe to these experiences [28,30]. Consequently, the findings of qualitative research tend to be reported in terms of descriptions and interpretations of the problem, which incorporate participants’ perspectives and the researcher’s reflexivity [28], rather than quantitatively measurable findings. 

Our results are not measurable in a quantitative sense – however, this does not mean the research findings are not substantial or meaningful. The identification of the different factors influencing farmers and technical services’ AMU behaviours as raised by the interviewees themselves, alongside an understanding of why these factors were important to the interviewees, are meaningful findings that could not have been generated using a quantitative approach. Furthermore, while qualitative research findings may not be statistically generalisable, they can be “theoretically recognisable” – that is to say, the “explanatory frameworks or typologies developed from the data, or mechanisms identified [can] have applicability beyond the immediate context of the reported study” [30]. In the case of this study, the specific barriers and facilitators to improving AMU practices for Indonesian poultry farmers that we identified are likely to be of relevance for other livestock farmers in South-East Asia, and can help indicate directions for future research hypotheses, for both quantitative and qualitative research.

Responses to comments on structure

Reviewer 1: Line 548, separate “study limitation” from “implication”, and write “Conclusion and recommendation” instead of implications as a new subheading.

This has been addressed in the manuscript (lines 571 and 635).

Reviewer 2: Lines 46-47: As it is different paragraph, complete the sentence……Interventions that promote responsible and prudent use of antimicrobials have the potential to mitigate the risk of AMR.

This has been addressed in the manuscript (lines 58-59).

Reviewer 2: Line 548: Conclusion should be separate

This has been addressed in the manuscript – there is now a “conclusion and recommendations” section (lines 635-644).

 

Bibliography

1. Anwar Sani R, Wagenaar JA, Dinar TEHA, Sunandar S, Nurbiyanti N, Suandy I, et al. The comparison and use of tools for quantification of antimicrobial use in Indonesian broiler farms. Frontiers in Veterinary Science [Internet]. 2023 [cited 2023 May 2];10. Available from: https://www.frontiersin.org/articles/10.3389/fvets.2023.1092302

2. Suandy I. AMU-AMR surveillance system in Indonesia on livestock and animal health sector: approach and findings. In Bangkok, Thailand; 2019. Available from: https://rr-asia.woah.org/wp-content/uploads/2020/01/11-amu-amr-surveillance-indonesia_saundy.pdf

3. Coyne L, Patrick I, Arief R, Benigno C, Kalpravidh W, McGrane J, et al. The Costs, Benefits and Human Behaviours for Antimicrobial Use in Small Commercial Broiler Chicken Systems in Indonesia. Antibiotics (Basel). 2020 Apr 1;9(4):E154. 

4. Usui M, Ozawa S, Onozato H, Kuge R, Obata Y, Uemae T, et al. Antimicrobial susceptibility of indicator bacteria isolated from chickens in Southeast Asian countries (Vietnam, Indonesia and Thailand). J Vet Med Sci. 2014 May;76(5):685–92. 

5. Takaichi M, Osawa K, Nomoto R, Nakanishi N, Kameoka M, Miura M, et al. Antibiotic Resistance in Non-Typhoidal Salmonella enterica Strains Isolated from Chicken Meat in Indonesia. Pathogens. 2022 May;11(5):543. 

6. Vasileiou K, Barnett J, Thorpe S, Young T. Characterising and justifying sample size sufficiency in interview-based studies: systematic analysis of qualitative health research over a 15-year period. BMC Med Res Methodol. 2018 Nov 21;18(1):148. 

7. Sandelowski M. Sample size in qualitative research. Research in Nursing & Health. 1995;18(2):179–83. 

8. Guest G, Arwen B, Johnson L. How Many Interviews Are Enough?: An Experiment with Data Saturation and Variability. Field Methods. 2006;18(1):59–82. 

9. Hennink M, Kaiser BN. Sample sizes for saturation in qualitative research: A systematic review of empirical tests. Social Science & Medicine. 2022 Jan 1;292:114523. 

10. Morse JM. Determining Sample Size. Qual Health Res. 2000 Jan 1;10(1):3–5. 

11. Malterud K, Siersma VD, Guassora AD. Sample Size in Qualitative Interview Studies: Guided by Information Power. Qual Health Res. 2016 Nov 1;26(13):1753–60. 

12. Carrique‐Mas JJ, Trung NV, Hoa NT, Mai HH, Thanh TH, Campbell JI, et al. Antimicrobial Usage in Chicken Production in the Mekong Delta of Vietnam. Zoonoses Public Health. 2015 Apr;62(s1):70–8. 

13. Imam T, Gibson JS, Foysal M, Das SB, Gupta SD, Fournié G, et al. A Cross-Sectional Study of Antimicrobial Usage on Commercial Broiler and Layer Chicken Farms in Bangladesh. Front Vet Sci. 2020;7:576113. 

14. Choisy M, Van Cuong N, Bao TD, Kiet BT, Hien BV, Thu HV, et al. Assessing antimicrobial misuse in small-scale chicken farms in Vietnam from an observational study. BMC Vet Res. 2019 Dec;15(1):206. 

15. Efendi R, Sudarnika E, Wibawan IWT, Purnawarman T. An assessment of knowledge and attitude toward antibiotic misuse by small-scale broiler farmers in Bogor, West Java, Indonesia. Vet World. 2022 Mar 25;707–13. 

16. Adam CJM, Fortané N, Coviglio A, Delesalle L, Ducrot C, Paul MC. Epidemiological assessment of the factors associated with antimicrobial use in French free-range broilers. BMC Vet Res. 2019 Jun 28;15(1):219. 

17. Christenson A, Johansson E, Reynisdottir S, Torgerson J, Hemmingsson E. Women’s Perceived Reasons for Their Excessive Postpartum Weight Retention: A Qualitative Interview Study. PLOS ONE. 2016 Dec 9;11(12):e0167731. 

18. Offord A, Turner H, Cooper M. Adolescent inpatient treatment for anorexia nervosa: a qualitative study exploring young adults’ retrospective views of treatment and discharge. European Eating Disorders Review. 2006;14(6):377–87. 

19. Rojo-Gimeno C, Dewulf J, Maes D, Wauters E. A systemic integrative framework to describe comprehensively a swine health system, Flanders as an example. Prev Vet Med. 2018 Jun 1;154:30–46. 

20. Doidge C, Hudson C, Lovatt F, Kaler J. To prescribe or not to prescribe? A factorial survey to explore veterinarians’ decision making when prescribing antimicrobials to sheep and beef farmers in the UK. PLOS ONE. 2019 Apr 9;14(4):e0213855. 

21. FAO. Preserving critically important antibiotics for humans, by banning their use in animals [Internet]. FAO in Indonesia. 2019 [cited 2023 Jul 13]. Available from: https://www.fao.org/indonesia/news/detail-events/en/c/1257265/

22. Charmaz K. Constructing grounded theory. London; Thousand Oaks, Calif: Sage Publications; 2006. 208 p. (Introducing Qualitative Methods). 

23. Braun V, Clarke V. Toward good practice in thematic analysis: Avoiding common problems and be(com)ing a knowing researcher. International Journal of Transgender Health. 2022 Oct 25;0(0):1–6. 

24. Braun V, Clarke V. Using thematic analysis in psychology. Qualitative Research in Psychology. 2006 Jan 1;3:77–101. 

25. Krouwel M, Jolly K, Greenfield S. Comparing Skype (video calling) and in-person qualitative interview modes in a study of people with irritable bowel syndrome – an exploratory comparative analysis. BMC Medical Research Methodology. 2019 Nov 29;19(1):219. 

26. Namey E, Guest G, O’Regan A, Godwin CL, Taylor J, Martinez A. How Does Mode of Qualitative Data Collection Affect Data and Cost? Findings from a Quasi-experimental Study. Field Methods. 2020 Feb;32(1):58–74. 

27. Pham-Duc P, Sriparamananthan K. Exploring gender differences in knowledge and practices related to antibiotic use in Southeast Asia: A scoping review. PLoS One. 2021;16(10):e0259069. 

28. Creswell JW. Qualitative inquiry and research design: Choosing among five approaches [Internet]. 2nd ed. 2007. Available from: https://www.researchgate.net/profile/Rulinawaty-Kasmad/publication/342229325_Second_Edition_QUALITATIVE_INQUIRY_RESEARCH_DESIGN_Choosing_Among_Five_Approaches/links/5eec7025458515814a6ac263/Second-Edition-QUALITATIVE-INQUIRY-RESEARCH-DESIGN-Choosing-Among-Five-Approaches.pdf

29. O’Brien BC, Harris IB, Beckman TJ, Reed DA, Cook DA. Standards for Reporting Qualitative Research: A Synthesis of Recommendations. Academic Medicine. 2014 Sep;89(9):1245. 

30. Barbour RS. The role of qualitative research in broadening the ‘evidence base’ for clinical practice. Journal of Evaluation in Clinical Practice. 2000;6(2):155–63.

---

## [Decision Letter · Decision Letter 1]

1 Sep 2023

“It's a habit. They've been doing it for decades and they feel good and safe.”: A qualitative study of barriers and opportunities to changing antimicrobial use in the Indonesian poultry sector

PONE-D-23-16108R1

Dear Dr. Hibbard

We’re pleased to inform you that your manuscript has been judged scientifically suitable for publication and will be formally accepted for publication once it meets all outstanding technical requirements.

Kind regards,

Sk Shaheenur Islam, DVM, MS, Ph.D.

Academic Editor

PLOS ONE

Additional Editor Comments (optional):

I greatly appreciate your tremendous efforts in revising the manuscript. Congratulations to all the authors for appropriately addressing the comments raised by the reviewers.

Reviewers' comments:

Reviewer's Responses to Questions

**Comments to the Author**

1. If the authors have adequately addressed your comments raised in a previous round of review and you feel that this manuscript is now acceptable for publication, you may indicate that here to bypass the “Comments to the Author” section, enter your conflict of interest statement in the “Confidential to Editor” section, and submit your "Accept" recommendation.

Reviewer #1: All comments have been addressed

Reviewer #2: All comments have been addressed

2. Is the manuscript technically sound, and do the data support the conclusions?

Reviewer #1: Yes

Reviewer #2: Yes

3. Has the statistical analysis been performed appropriately and rigorously? 

Reviewer #1: I Don't Know

Reviewer #2: I Don't Know

4. Have the authors made all data underlying the findings in their manuscript fully available?

Reviewer #1: Yes

Reviewer #2: Yes

5. Is the manuscript presented in an intelligible fashion and written in standard English?

Reviewer #1: No

Reviewer #2: Yes

6. Review Comments to the Author

Reviewer #1: Title: hey've been doing it for decades and they feel good and safe.”: A qualitative study of barriers and opportunities to changing antimicrobial use in the Indonesian poultry sector

This is a nice study. Thanks for addressing the comments p[roperly.

Reviewer #2: Thank you for your tremendous efforts to update the manuscript. Congratulations to all authors to accept the manuscript at Plos One

7. PLOS authors have the option to publish the peer review history of their article (what does this mean?). If published, this will include your full peer review and any attached files.

Reviewer #1: **Yes: **Prof. Dr. Md. Tanvir Rahman

Reviewer #2: No

---

## [Editor Report · Acceptance letter]

14 Sep 2023

PONE-D-23-16108R1 

“It's a habit. They've been doing it for decades and they feel good and safe.”: A qualitative study of barriers and opportunities to changing antimicrobial use in the Indonesian poultry sector 

Dear Dr. Hibbard:

I'm pleased to inform you that your manuscript has been deemed suitable for publication in PLOS ONE. Congratulations! Your manuscript is now with our production department. 

Kind regards, 

on behalf of

Dr. Sk Shaheenur Islam 

Academic Editor

PLOS ONE